# On the Compressive Response of Polymeric Cellular Materials

**DOI:** 10.3390/ma13020457

**Published:** 2020-01-18

**Authors:** Stefano Del Rosso, Lorenzo Iannucci

**Affiliations:** Department of Aeronautics, Imperial College London, London SW7 2AZ, UK; l.iannucci@imperial.ac.uk

**Keywords:** foams, 3D printing, dynamic compression tests

## Abstract

This paper presents a series of compression tests performed on a variety of high performance lightweight cellular materials conventionally used in energy absorption applications. Compressive tests were performed over a range of strain rates with a universal testing machine and a single stage gas gun. Experimental results revealed a dependency of the mechanical properties on the polymeric precursor, density, infill topology and strain rates. The dynamic strength of the investigated materials was determined through a material parameter identification study via the finite element (FE) method. Numerical results matched well with the experimental results and revealed a substantial enhancement in the compressive strength of the tested material from quasi-static to dynamic loading regimes by as much as 87%. The strength of 3D printed polymers was superior with respect to the tested polymeric foams. On the other hand, polymeric foams showed higher efficiency and energy absorption ability.

## 1. Introduction

Cellular materials such as honeycombs and foams are increasingly exploited in engineering applications due to their excellent ability to absorb energy, low weight and relatively low cost. The main applications for soft and semi-rigid foams include insulation, packaging and cushioning, whilst rigid foams and honeycombs are also employed in structural applications, as well as the core of lightweight sandwich structures. The selection of the material is application dependant and should be based on factors such as in-service force and pressure, duration and direction of the load, design constraints, environmental conditions and cost. Hence, a thorough understanding and evaluation of the mechanical behaviour under different loading conditions is necessary in order to determine the range of applications in which certain materials would have the best response. There is extensive experimental data available in the open literature on the mechanical properties of polymeric foams. For example, different peer reviewed articles document the compressive properties of polyurethane foams [1,2,3,4], expanded polystyrene foams [4,5,6], expanded polypropylene foams [6,7], polyethylene foams [4,6], polyvinyl chloride foams [8,9] and polymethacrylimide foams [10,11,12,13].

It is generally accepted and was experimentally confirmed in [4,5,6,7,8,9,10,13,14] that polymeric foams exhibit basic mechanical properties and temperature and strain rate dependences, which are derived from the characteristics of the precursor solid material. The collapse strength $\sigma_{Y}$ and plateau strength $\sigma_{PL}$ increase with increasing density ρ and strain rate $\dot{\varepsilon }$, whilst densification occurs at lower strains [14,15].

In the last 30 years, additive manufacturing (AM) technology has been rapidly developed and is increasingly exploited for the creation of prototypes and complex shape parts. Thermoplastic polymeric materials such as ABS, PA, PLA and TPU can be processed using such techniques [16]. When the raw material is transformed into a part with an infill percentage less than 100% during fused filament fabrication, it is possible to classify the printed part as a cellular solid. The mechanical properties and energy absorption ability of monolithic and lattice 3D printed structures have been documented in numerous studies. For example, Tagarielli et al. [17] investigated the mechanical properties of 3D printed PLA, highlighting the mechanical anisotropy resulting from the printing direction. Nevertheless, 3D printed PLA was noted to be tougher, more crystalline and less sensitive to the strain rate with respect injection moulded PLA. Bates et al. [18] investigated the energy absorption ability of 3D printed TPU honeycombs with different relative densities and printing orientations. The authors found that the fabricated structures had comparable energy absorbing efficiency to commercially available expanded closed cell polyurethane foams. The honeycomb structures were able to recover their initial shape elastically after the removal of loads. However, a detrimental effect in the energy absorption ability was observed when the TPU structures were cyclically loaded. Habib et al. [19] investigated the compressive behaviour of 3D printed PA honeycombs with different wall thicknesses and printing directions. The authors found excellent agreement of the experimental results with the analytical methods conventionally used for the prediction of the mechanical properties of metal honeycombs.

One of the fields in which AM could have enormous potential is in the construction of nano and micro Unmanned Aerial Systems (UASs). Yeong et al. [20] reviewed the current state-of-the-art of such technology applied to the manufacturing of UASs, highlighting benefits and limitations. Current nano and micro UASs use foam or carbon fibre reinforced polymer (CFRP) protections in order to limit the damage of structural parts to low velocity impacts [21]. However, these designs offer no protection and insignificant levels of energy dissipation in the case of impacts with manned aircraft at take-off, cruising and landing velocities. It has been shown via numerical predictions that the pressures generated by UAS components upon impact can be very localised and higher than pressures generated during a typical bird impact [22]. In 2016, the FAA reported an increased number of drone sightings from manned aircraft pilots with respect to the previous year [23]. It is envisaged that future drone designs will take into account the possibility of high energetic impact scenarios in order to mitigate the severity of the impact and lower peak energies and pressures.

This paper presents a series of compression tests performed on a range of lightweight materials conventionally used in energy absorption applications and 3D printed cellular solids. The experimental programme consisted of quasi-static compression tests performed with three different strain rates on thirty six different types of materials. The materials that showed the highest compressive strength, sensitivity to the strain rate, absorbed energy and efficiency were tested under dynamic conditions similar to a Taylor impact test. Hence, the finite element (FE) method was employed in order to quantify the strength at higher strain rates. The FE methodology involved the solution of an inverse problem where the material parameters were calibrated by matching the response of the virtual specimen with the experimentally observed response.

## 2. Materials, Manufacturing and Testing Method

### 2.1. Materials

A series of commercially available lightweight materials were procured and tested. The nominal thickness of the as-received panels was 25 mm. The physical properties of the investigated materials are listed in Table 1. All foams had a closed-cell structure.

3D printed specimens were prepared in-house using two different 3D printers (Makerbot Replicator 2X and Ultimaker 3 Extended) via the fused filament fabrication (FFF) method. A variety of raw materials (ABS, CPE, PA and PLA) and different infill densities (10% and 30%) and infill topologies (linear, triangular and hexagonal) were investigated. Table 2 presents the physical properties of the 3D printed specimens. All raw materials were processed using the recommended temperature settings (Table 3). Cross-sections of the 3D printed specimens are shown in Figure 1. For all specimens, the faces of the infill shells were perpendicular to the direction of the build (out-of-plane). Throughout the paper, the investigated materials will be referenced by the assigned ID code.

### 2.2. Testing Methods

The experimental testing campaign was divided into two levels of testing. Level 1 consisted of conventional quasi-static compression tests on all materials listed in Table 1 and Table 2. The materials that showed the best combination of collapse strength, energy absorption, efficiency and sensitivity to the strain rate were tested under dynamic conditions (Level 2).

#### 2.2.1. Quasi-Static Compression Tests

Uniaxial compression tests were carried out according to ASTM D1621-16 “Standard Test Method for Compressive Properties of Rigid Cellular Plastics” [24]. Tests at quasi-static regimes were performed using a universal testing machine (Instron 5969) equipped with a 50 kN load cell with an accuracy of ±0.5% of the displayed force (Figure 2). Square test specimens of nominal 50 mm × 50 mm dimensions were cut from the as-supplied materials using a wire saw. The specimens were placed between the compression platens ensuring that the specimen centreline was aligned with the load cell centreline. In order to investigate the effect of the strain rate, compressive tests were performed at three different cross-head speeds of 2 mm/min, 20 mm/min and 200 mm/min, corresponding to a strain rate of 0.001 s^−1^, 0.013 s^−1^ and 0.133 s^−1^, respectively. All tests were performed at room temperature. Quasi-static compression tests on 3D printed specimens were performed with cylinders of 20 mm diameter and 25 mm long. Although not recommended by the standard, this specimen geometry was chosen in order to compare the compressive properties of the specimens under quasi-static and dynamic conditions directly. It was necessary to print the skin to ensure the stability of the specimen and replicate periodic boundaries as in real structures.

The compressive stress was determined by dividing the load by the initial cross-sectional area of the specimen;As cellular materials experience variations in the local strain field under compression, the average compressive strain was determined optically by tracking the relative displacement of two points drawn on the compressive platens.

#### 2.2.2. Dynamic Compression Tests

The dynamic characterisation of selected materials was carried out by firing the specimens onto a rigid bar anvil, in the manner of a Taylor test [25], using a 50 mm calibre single stage gas gun. Figure 3 shows the typical test setup. The gas gun had two 10 lpressure vessels in which the launch gas could be pressurised up to 330 bar. The stainless steel bar was 2 m long with a diameter of 30 mm and horizontally supported by rolling bearings. Cylindrical specimens of 20 mm in diameter and 25 mm long were bonded on top of the sabot using double sided tape (3M 300LSE) and inserted into the breech of the gas gun. The breech plug was inserted and screwed closed, pushing the sabot in the firing position. The capture tank was closed and vacuum drawn. Once the vacuum level reached 100 mBar, the gun was pressurised to the desired firing pressure. The sabot/projectile system was accelerated along the 5 m long smooth barrel. At the end of the barrel, a “sabot-stripper” stopped the sabot, allowing only the specimen to fly further onto the target. All specimens were fired at a nominal impact velocity of 160 m/s. A high speed camera (Phantom v2512) was positioned in front of the open window normal to the tip of the bar in order to record the impact event. The camera was equipped with a Zeiss 100 mm Makro-planar lens and was set to record 601,850 frames per second (fps) with a resolution of 128 × 64 pixels and exposure of 1.125 μs. Black dots were marked on the surface of the specimens with a felt pen, and their position was tracked during the test. Videos were post-processed using Imetrum^®^ Video Gauge^TM^ software. The diameter of the specimen was chosen such that the cross-sectional area of the specimen was smaller with respect to the cross-sectional area of the tip of the Taylor bar. The length of the foam specimens was chosen to be the same as the thickness of the received billets in order to avoid cuts and bonding with adhesives. The length of 3D printed specimens was chosen to be the same as the other specimens in order to have a direct comparison throughout the testing campaign. The tip of the bar was lubricated with silicone grease to minimise frictional effects. The effect of the vacuum during the tests was neglected. For such testing conditions, the compressibility of the air would be insignificant compared to the structural stiffness of the tested materials.

## 3. Results and Discussion

### 3.1. Quasi-Static Compression Tests

Figure 4 shows the compression stress vs. strain curves for the investigated foams tested at different quasi-static strain rates. Only one curve per material is shown for clarity purposes.

It could be observed that the global compressive behaviour of foams depended mainly on their precursor polymer and its relative density and the mechanical properties related to the cell structure. Generally, the higher the density, the higher the collapse stress $\sigma_{Y}$ and the lower the densification initiation strain $\varepsilon_{d0}$. The initial (elastic) part of the stress vs. strain curve of rigid and semi-rigid foams was controlled by the cell edge and wall bending with stiffness proportional to the relative density. After exceeding the collapse strength $\sigma_{Y}$, the long plastic plateau could be associated with the collapse of the cells (plastic buckling and brittle crushing). In this region, the stress $\sigma_{PL}$ remained fairly constant. Densification occurred when all the cells locked up. This point was marked by a steep increase in stress with increasing strain. The compressive stress vs. strain behaviour of soft foams did not show a clearly defined yield point with $\sigma_{Y}$ and $\sigma_{PL}$ between 100 kPa and 400 kPa and densification strains greater than 70%. Elastic buckling occurred up to $\varepsilon_{d0}$, where the stress increased steeply [14].

The highest collapse strength $\sigma_{Y}$ was noted for rigid foams having the highest density (*I, L, O, R*) and ranged between 2.6 MPa and 5.3 MPa when tested at the slowest cross-head displacement rate. Soft and semi-rigid foams were the most compliant with a $\sigma_{Y}$, arbitrarily defined as the stress at 10% strain, below 0.5 MPa. All tested materials showed a strain rate dependency with PMI foams showing the highest increase in $\sigma_{Y}$ (up to 23%) and EPS foams the smallest (up to 6%) across the investigated quasi-static conditions.

Soft foams (Types *F, G, S, T*) almost completely recovered their initial shape after removing the compressive load, regardless of the applied strain rate. As their structure was more compliant, low density PE and PVDF based foams are not used for structural applications, but mainly for their electro-magnetic and thermal properties, as well as acoustic insulation [26]. Foam Types *A, B, C, D, E, J, K, L* plastically deformed with a partial recovery of their initial shape after removing the compressive load. Foam Types *H, I, M, N, O, P, Q, R* plastically deformed without any shape recovery. Instead, small crumbles were observed on the surfaces of the latter types, indicating brittle crushing.

Figure 5 shows the compression stress vs. strain curves for the investigated 3D printed specimens tested at different quasi-static strain rates. The initial elastic compressive behaviour of 3D printed materials mainly depended on the precursor polymer and the infill density. However, contrary to what was observed while testing foams, in which the crush plateau was smooth and approximately stable at a value of $\sigma_{PL}$, the plateau region for 3D printed specimens showed features similar to those observed for aluminium tubes tested in compression [27,28]. The stress vs. strain curves revealed smooth peaks and troughs, which could be associated with the folding of the cylindrical structure in a concertina mode. This phenomenon was more evident in stiffer and denser specimens than in compliant ones. Amongst the investigated materials, PLA had the highest collapse strength, which was calculated between 23 MPa and 27 MPa, depending on the test rate, for the specimens having a 30% infill density, followed by ABS, PA and CPE. The influence of the infill topology was minimal as far as $\sigma_{Y}$ was concerned. However, it was observed that specimens with a linear infill pattern had a more stable and longer yield with respect to specimens with a triangular pattern, which showed an earlier softening after yielding and a lower densification initiation strain for the same infill percentage. 3D printed specimens showed less sensitivity to the strain rate than foams, with $\sigma_{Y}$ increasing by as much as 14% for the investigated quasi-static rates. All 3D printed specimens plastically deformed upon compression. PA specimens were the only ones able to recover their initial cylindrical shape partially after removing the applied load. The lesser the infill density, the greater the shape recovery. Representative images of pristine and tested specimens are shown in Figure 6.

### 3.2. Energy Absorption and Efficiency Diagrams

It is possible to analyse the quasi-static behaviour of the tested materials by means of the energy absorption and efficiency diagrams. The absorbed energy W is defined as the area under the stress vs. strain curve (Equation (Equation 1)). The efficiency parameter η is defined as the ratio between the absorbed energy up to the stress σ and the stress itself (Equation (Equation 2)): (1)$$W=\int_{0}^{\varepsilon }\sigma \left(\varepsilon \right)d\varepsilon$$
(2)$$\eta \left(\varepsilon \right)=\frac{W}{\sigma (\varepsilon )}$$

By plotting W vs. σ and η vs. σ, it was possible to identify which material would perform optimally given an amount of energy to be absorbed, stress and strain. Figure 7 shows representative energy absorbed and efficiency plots for two different materials. The trends observed for Types *A, B, C* and *AC, AD* were similar to those observed for the other investigated materials.

It can be seen from Figure 7a,c that, for a given material density, the energy absorption increased as the stress and strain rate increased. The envelope would represent optimum energy absorption levels for materials of any density [15]. The envelope gradient could also be interpreted as the maximum amount of energy that a material could absorb up to $\varepsilon_{d0}$. The higher the slope, the higher the amount of energy. Efficiency diagrams, as shown in Figure 7b,d, are another way to visualise the ability of a material with a known density to absorb energy. They can be also used to objectively define the densification initiation strain $\varepsilon_{d0}$ as the strain when η(ε) reaches its maximum. Figure 8 shows the envelope gradients and the efficiency parameters for the investigated materials. From this plot, it can be deduced that, amongst the investigated cellular materials, PMI-based foams appeared to be the best types of foams in terms of the ability to absorb energy, in particular Rohacell^®^ RIST and Rohacell^®^ IG. EPS foams had an envelope gradient 14% and 6% lower with respect to the former materials, respectively. The envelope gradient calculated for 3D printed PLA with a linear infill pattern was 0.505, i.e., the highest amongst the 3D printed materials. Nevertheless, its gradient was about 16% smaller with respect to Rohacell^®^ RIST. The lowest ability to absorb energy was found for 3D printed CPE specimens, which were brittle and compliant. It also appears from Figure 8 that 3D printed specimens with a linear infill pattern were superior with respect to those having a triangular infill. As previously mentioned, the plateau region of the former 3D printed specimens was steadier with respect to stress in the same region noted for the specimens having a linear infill architecture. The difference in efficiency and envelope gradient between linear and hexagonal infill pattern was within the scatter error bars. However, further tests and characterisations are needed in order to understand the differences observed between the three patterns thoroughly and confirm the conclusions drawn.

A summary of the collapse stress vs. density is shown in Figure 9. Comparing the envelope gradients in Figure 7 and the trends observed in Figure 9, it is possible to reiterate the fact that PMI-based foams and PLA-based 3D printed specimens would be the most suitable materials for energy absorption applications. The selection of the material density would be application dependent and based on the experienced in-service force and energy thresholds.

### 3.3. Dynamic Compression Tests

Although PLA specimens had the highest collapse strength amongst the investigated materials, the plateau stress was unsteady and, when normalised with respect to the specimen density, similar to the normalised plateau stress of ABS specimens. Hence, the latter material was selected for dynamic compression tests.

Figure 10 shows a series of snapshots taken while testing selected specimens under dynamic strain rates. Figure 11a,b shows the instantaneous velocity vs. time history and the crush strain vs. time history, respectively, of the tested specimens impacting on the rigid steel bar.

Upon impact, 3D printed ABS specimen Types *U* and *W* progressively plastically buckled in a similar way to axially compressed tubes, whilst 30% filled specimens showed plastic deformation only near the impact face, which mushroomed. It also appeared that the 3D printed specimens with the highest core density produced the steepest reduction in velocity (Figure 11a). Conversely, the most compliant tested material (Type *G*, Plastazote^®^ HD60) showed the slowest deceleration and the highest compression strain amongst the investigated ones. The impact was almost completely elastic, and rebound velocity was about 42% compared with the initial velocity. Fluctuations in the rebound velocity could be attributed to elastic waves bouncing within the specimen itself. Amongst the tested materials, Type *G* was the only one able to recover its initial shape after impact. PET and PMI based foams pulverised upon impact at the impact face (Figure 10b,c, respectively), where the load was highly localised and micro-inertia phenomena developed [14]. These foams experienced global compression strains between 30% and 40%. It was not possible to recover fragments for post-mortem inspections. Note that all tested foams were isotropic, whilst the 3D printed materials had a strong orthotropic nature due to their build and, hence, could be tailored. However, the design and optimisation of the infill topology are beyond on the scope of this study. The compression of the materials under dynamic conditions followed a step-wise loading approach (observed within the first 100 μs after impact). The low elastic and plastic (crush) wave speeds in the foams meant the specimens were not in dynamic equilibrium, and hence, step-wise compression was noted. This phenomenon was not observed during static compression tests.

### 3.4. Understanding the Dynamic Impact via the Finite Element Method

Although the Taylor bar was equipped with a dynamic load cell (Strainsense 10 kN Annular Load Cell) mounted behind the tip of the bar, it was not possible to record any meaningful data. The momentum generated upon impact was too low to trigger the load cell, regardless of the set force threshold. Therefore, the dynamic strengths of the tested materials were calculated through a finite element (FE) study. FE can also simplify the problem as variations of specimen density, inertia effects and instantaneous acceleration are automatically computed by the solver.

The simulations of the dynamic compression tests were carried out using the explicit LS-Dyna FE code [29]. For each foam type, a parameter identification study was carried out to find the best input to match the experimental crush deformation and the velocity reduction in Figure 11. For foams, cylindrical specimens with a 10 mm radius and 25 mm long were modelled using brick solid elements (*ELFORM= 1; element size = 1 mm^3^; element aspect ratio between one and 1.8). The virtual specimens were set to impact a planar rigid wall with an initial velocity equal to 160 m/s. No friction was defined upon the contact between the specimen and the rigid wall.

Among the different material models available in the FE solver, the Material Type 63 *MAT_CRUSHABLE_FOAM was selected for all the simulations of foam specimens. *MAT_CRUSHABLE_FOAM requires the user to input five parameters: density (RO), elastic modulus (E), Poisson’s ratio (PR), a load curve LCID, which defines the yield stress vs. volumetric strain, tensile stress cut-off (TSC) and a rate sensitivity damping coefficient (DAMP). Nevertheless, when an LCID was defined, it was found that the variation of E and TSC did not affect the behaviour of the virtual specimen. For all simulations, the values of PR and DAMP were set to zero and 0.1, respectively, whilst the value of RO was equal to the density of the real specimen. A piecewise stress vs. volumetric strain curve (Figure 12a) was used to idealise the mechanical behaviour of the specimen. The values of X${}_{1}$ and Y${}_{1}$ were the only variables investigated in the study. For the foams that crumbled upon impact onto the Taylor bar, an additional material card (*MAT_ADD_EROSION) was used. This card was set to remove elements when the principal stress in those elements reached the value of X${}_{1}$. In order to compare the experimental results with those obtained from the FE simulations, the velocity and the distance of a “node of interest”, the Z-coordinate of which was equal to the Z-coordinate of the specimen’s “point of interest”, were analysed (Figure 12b).

The FE models of 3D printed specimens were created so as to reflect the actual geometry and topology of their real counterparts. Figure 13 shows the sections of the FE models. Shell elements (*ELFORM = 1; element size = 1 mm^2^; element aspect ratio between one and 2.2) with a thickness of 0.3 mm and 0.6 mm were used for the core and for the outer skin, respectively. The impact conditions were the same as those used for the crash simulations of the foams. Amongst the different material cards available in LS-Dyna, an elasto-plastic constitutive material model (Material Type 24) was chosen. *MAT_24 requires the user to define an arbitrary stress vs. strain curve via the definition of effective plastic strain values and their corresponding yield stress values. In order to calibrate the models, the Young’s modulus E and the yield strength SIGY were set as variables. The plateau stress was assumed constant at the value of $\sigma_{Y}$, and the strain to failure was set equal to 8% for all the simulations. In order to simulate erosion, the *MAT_ADD_EROSION card was used and controlled by the effective plastic strain (EFFEPS).

#### FE Results

As mentioned in the previous sections, an inverse approach was adopted to determine the dynamic strength of the specimens via the FE method. The material parameters were iteratively varied until a close correlation between the numerical and experimental results was found. In order to establish the predictive ability of the material model used, quasi-static compression tests were simulated using the stress vs. strain relationship sketched in Figure 12a and values of X${}_{1}$ and Y${}_{1}$ as collected during the experimental campaign. A comparison between the FE simulations and experimental results for specimen Type *O* (Rohacell^®^ 110RIST) is shown in Figure 14.

The correlation between numerical and experimental results for quasi-static tests was good by using a simple elastic perfectly plastic stress vs. strain relationship with a Young’s modulus equal to 160 MPa for all the simulations and Y${}_{1}$ values equal to 2.83 MPa, 3.2 MPa and 3.45 MPa for cross-head displacement rates of 20 mm/min, 20 mm/min and 200 mm/min, respectively.

Once the validity of the numerical model was established, simulations were attempted for the dynamic cases. Figure 15 shows a comparison between the experiment and the numerical simulations for foam Type *L* (Rohacell^®^ 110RIMA) under dynamic compression. The same approach was adopted for all the other impact cases (Figure 16a,b). High values of $\sigma_{pl}$ allowed quicker initial velocity reductions and smaller specimens deformations, whilst smaller $\sigma_{pl}$ values led to more compliant responses. The best match was found when using plateau stress values between 5.2 MPa and 7.0 MPa. Variations of X${}_{1}$ between 0.005 and 0.04 did not significantly affect the initial response of the virtual specimen, but only the rebound velocity. Comparing the snapshots of the simulation and experiment in Figure 15c, it is possible to note a small discrepancy in the specimen length, which increased with time. The virtual specimen appeared to recover its initial geometry at a faster rate compared to the real specimen. In order to capture recoverable plastic deformations, a more refined stress vs. strain behaviour should be used. Moreover, more advanced material cards, such as MAT_FU_CHANG (MAT_83) or MAT_MODIFIED_CRUSHABLE_FOAM (MAT_163), could also be used to obtain a closer correlation with this particular type of foam. Nevertheless, MAT_63 showed an excellent ability to capture the compressive behaviour of a variety of foams [29].

Figure 16c,d presents the comparison between the experimental and numerical results for 3D printed specimens tested under dynamic conditions. Even for these types of cellular structures, a good correlation was found through the inverse parameter identification approach. Nevertheless, small differences could be observed especially for low infill density specimens. The reason for the mismatch could be attributed to a few factors such as geometrical discrepancies between the model and the real counterpart, non-uniform local density and the lack of model strain rate dependency. The use of a more complex material card, in which parameters such as transverse and shear properties can be defined, would help to refine the model and find a better correlation with the experimental results.

The values of the plateau stress used for the simulations are plotted in Figure 17 along the experimental values. It is possible to see that the plateau stress values increased with increasing the test speed. The enhancement in $\sigma_{pl}$ at 10% strain was quantified between 4% and 87%. A linear relationship between the plateau stress values and the logarithm of the test speed for all the investigated materials also appeared to exist. Although the R-squared fit values for the foams were close to unity, a worse correlation was found for the 3D printed materials. This could be associated with the higher variability in the experimental test data along the plateau region, which was not constant up to the densification initiation strain. These results were in agreement with results reported in the literature. For example, Ibrahim [30] noted a 40% increase of the plateau stress of PMI foams when the test speed increased from 0.127 mm/s to 12.7 mm/s. Krundaeva et al. [31] noted an increase of $\sigma_{pl}$ of about 70% for EPS foams tested at strain rates between 10 s^−1^ and 100 s^−1^. Strength enhancement was found in specimens 3D printed with ABS when tested in tension under strain rates between 0.0127 mm/min and 100 mm/min [32] or in compression [33].

## 4. Conclusions

In this work, the compressive behaviour of various lightweight materials (polymeric foams and 3D printed polymers) was experimentally investigated through a series of mechanical tests. The investigated materials were subjected to different loading conditions, from quasi-static regimes to impact loading. The experimental results revealed marked differences in the compressive response of the investigated material as far as the stress and strain were concerned. All the investigated materials were strain rate sensitive and showed enhanced strength with an increase in strain rate.

Amongst the thirty six tested materials, ten were selected for the dynamic characterisation under dynamic conditions. A single stage gas gun was employed to fire cylindrical specimens onto a rigid bar anvil, in the manner of a Taylor test. High speed cameras were employed to record the deformation history of the specimen. The experimental results revealed an increase of the compressive strength of the materials with increasing the test strain rate. The mechanical properties were dependant on the material density, precursor material and infill topology. It was found that 3D printed materials can have higher collapse stress with respect to polymeric foams. On the other hand, the latter would be more efficient and able to absorb higher amounts of energy compared to the former class of materials for the same density. In order to determine the strength of the materials under dynamic conditions, the FE method was used. Numerical results matched well with the experimental results and revealed a further increase of the plateau stress by as much as 87% with respect to the strength calculated at quasi-static test rates.

## Figures and Tables

**Figure 1 materials-13-00457-f001:**
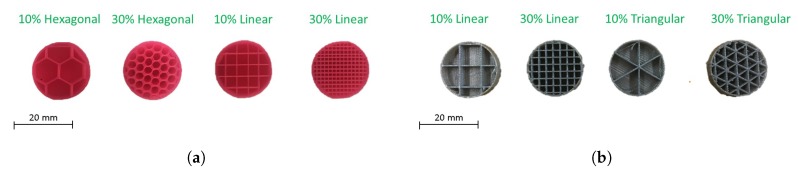
Cross-section of 3D printed specimens manufactured with: (**a**) Makerbot Replicator 2X; (**b**) Ultimaker 3 Extended.

**Figure 2 materials-13-00457-f002:**
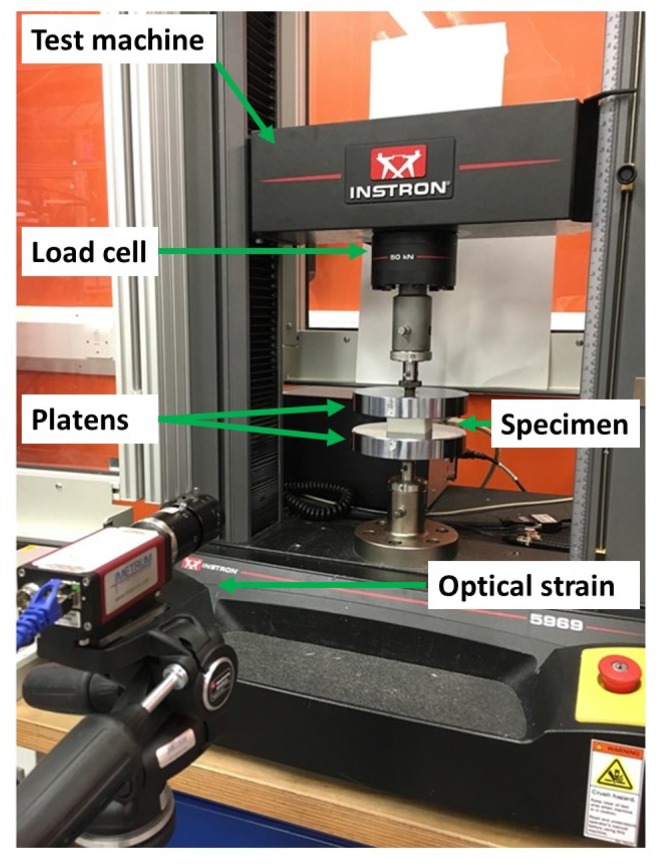
Quasi-static compression test setup.

**Figure 3 materials-13-00457-f003:**
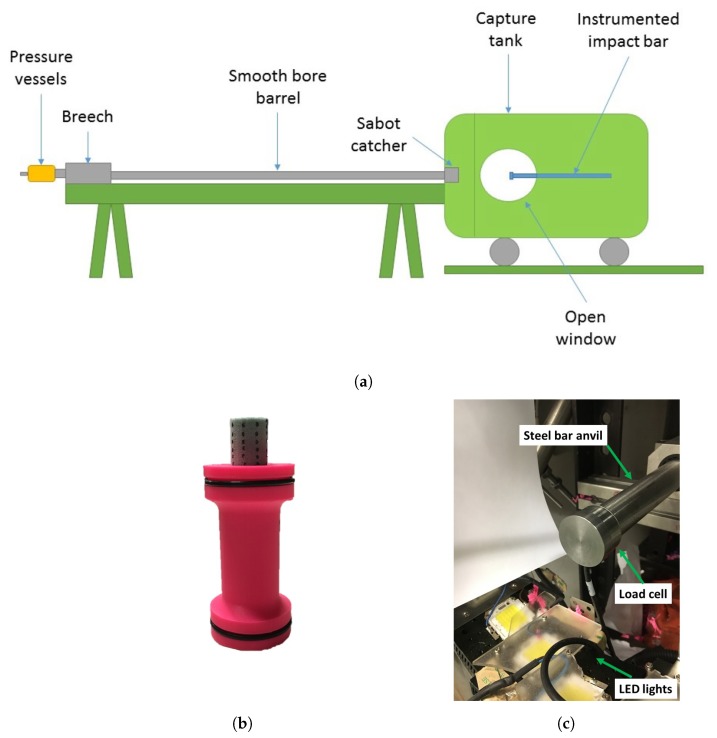
(**a**) Imperial College 50 mm gas gun; (**b**) sabot/specimen system for dynamic compression tests; (**c**) tip of the impacted steel bar.

**Figure 4 materials-13-00457-f004:**
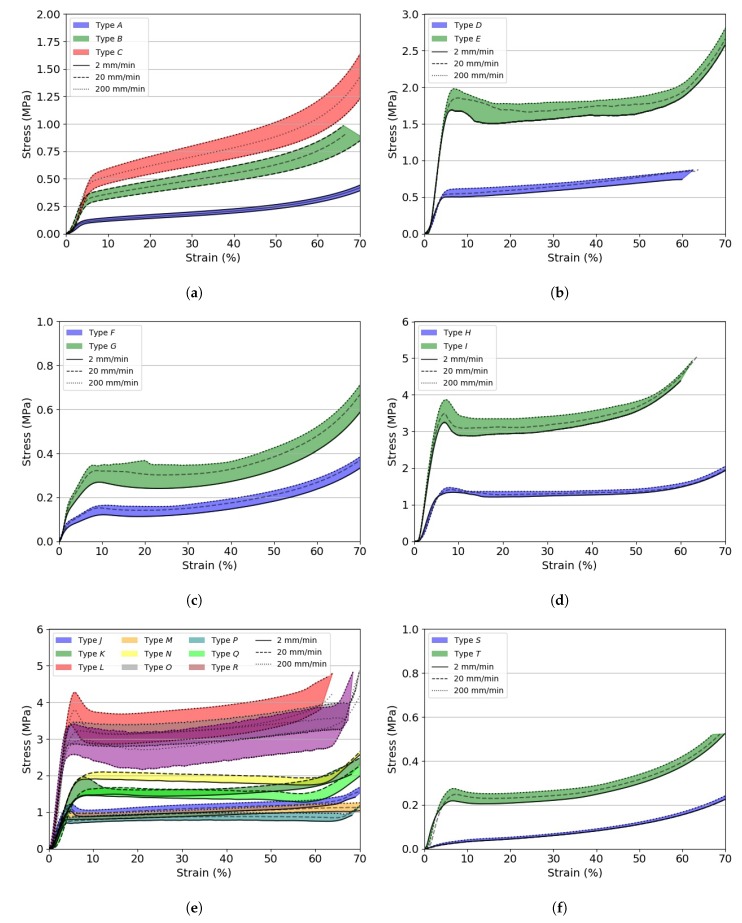
Compressive stress vs. strain curves for the investigated foams tested at different cross-head displacement rates. (**a**) EPP foams; (**b**) EPS foams; (**c**) PE foams; (**d**) PET foams; (**e**) PMI foams; (**f**) PVDF foams.

**Figure 5 materials-13-00457-f005:**
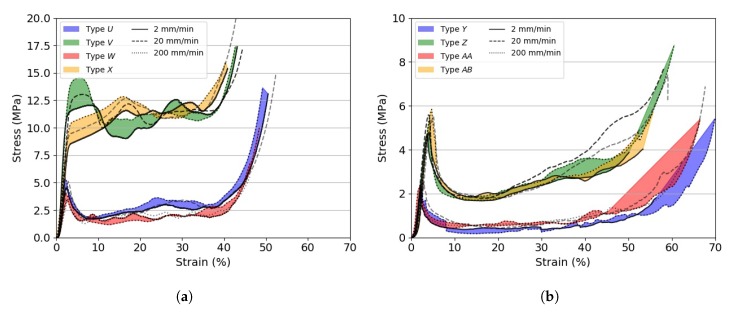
Compressive stress vs. strain curves for the investigated 3D printed specimens tested at different cross-head displacement rates. (**a**) 3D printed ABS; (**b**) 3D printed CPE; (**c**) 3D printed PA; (**d**) 3D printed PLA.

**Figure 6 materials-13-00457-f006:**
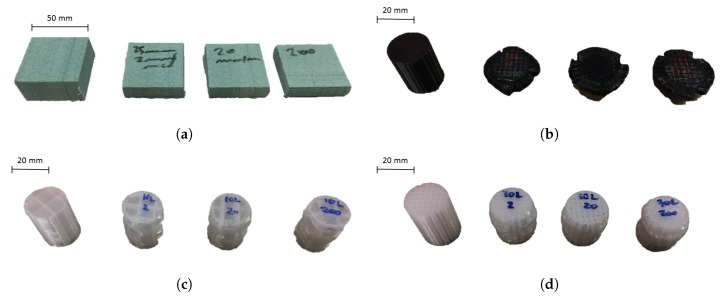
Representative specimens tested under quasi-static conditions. From left to right: pristine, 2 mm/min, 20 mm/min and 200 mm/min. (**a**) Type I (PET); (**b**) Type AH (3D printed PLA); (**c**) Type AC (3D printed PA); (**d**) Type AD (3D printed PA).

**Figure 7 materials-13-00457-f007:**
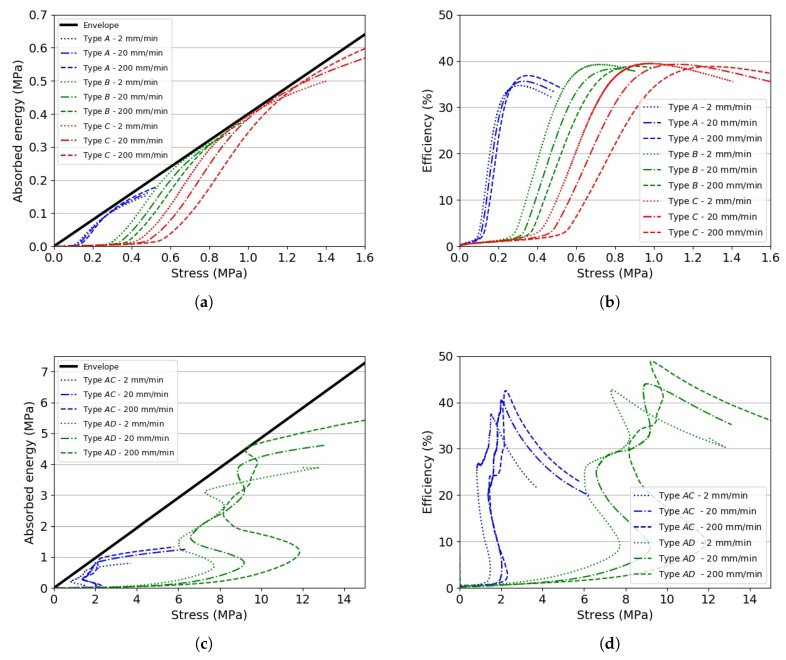
(**a**) Absorbed energy vs. stress for Specimens A, B and C (EPP); (**b**) efficiency diagram for Specimens A, B and C (EPP); (**c**) absorbed energy vs. stress for Specimens AC and AD (3D printed PA); (**d**) efficiency diagram for Specimens AC and AD (3D printed PA).

**Figure 8 materials-13-00457-f008:**
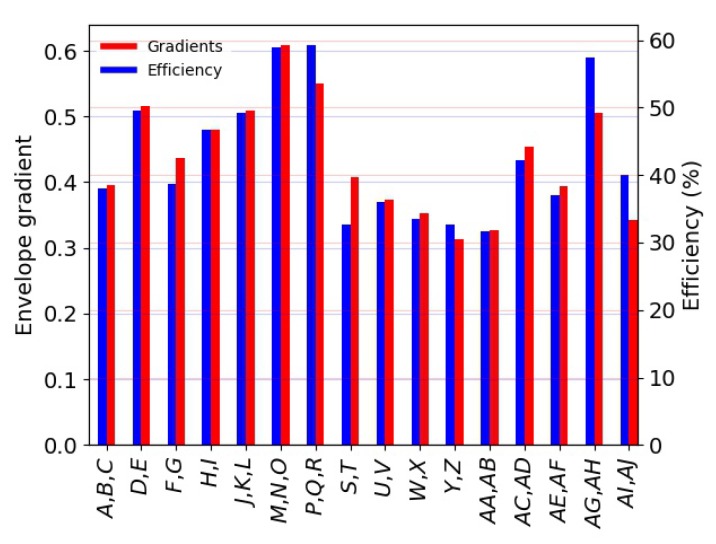
Envelope gradient and efficiency parameter for the investigated materials.

**Figure 9 materials-13-00457-f009:**
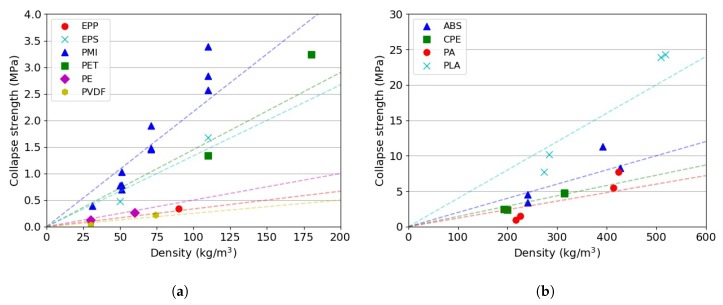
Collapse strength vs. density for the investigated materials (data from 2 mm/min compression tests). (**a**) Foams; (**b**) 3D printed.

**Figure 10 materials-13-00457-f010:**
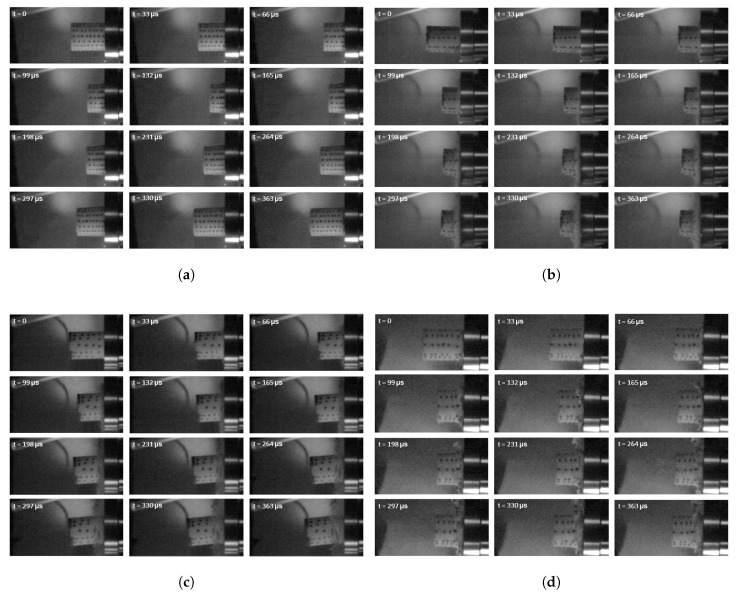
Series of snapshots from dynamic compression tests for selected materials. (**a**) Type G (Plastazote^®^ HD60); (**b**) Type I (PET); (**c**) Type O (Rohacell^®^ 110RIST); (**d**) Type W (3D printed ABS).

**Figure 11 materials-13-00457-f011:**
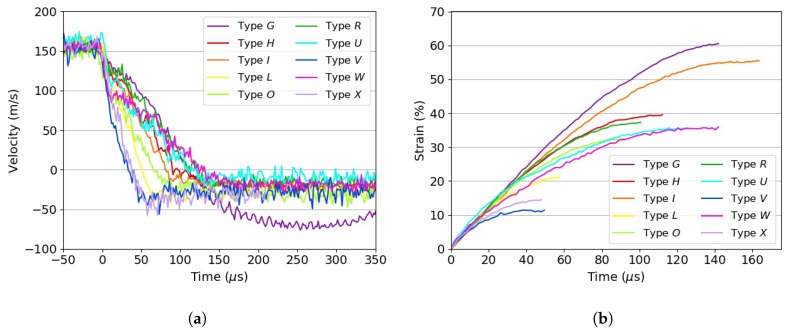
Dynamic test results for the investigated materials. (**a**) Velocity vs. time history; (**b**) strain vs. time history.

**Figure 12 materials-13-00457-f012:**
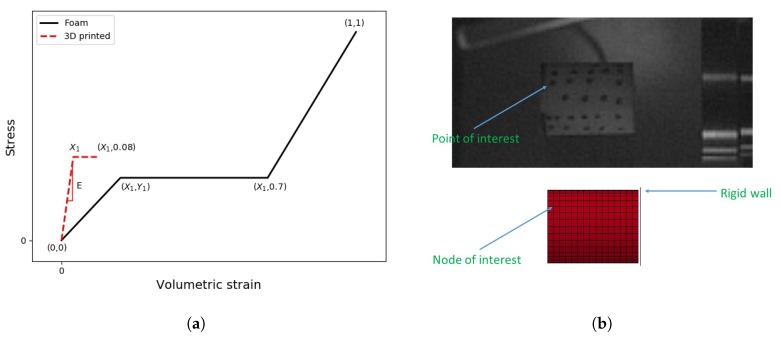
(**a**) Stress vs. volumetric strain of a virtual foam specimen; (**b**) definition of the points of interest for the experiment and the FE model.

**Figure 13 materials-13-00457-f013:**
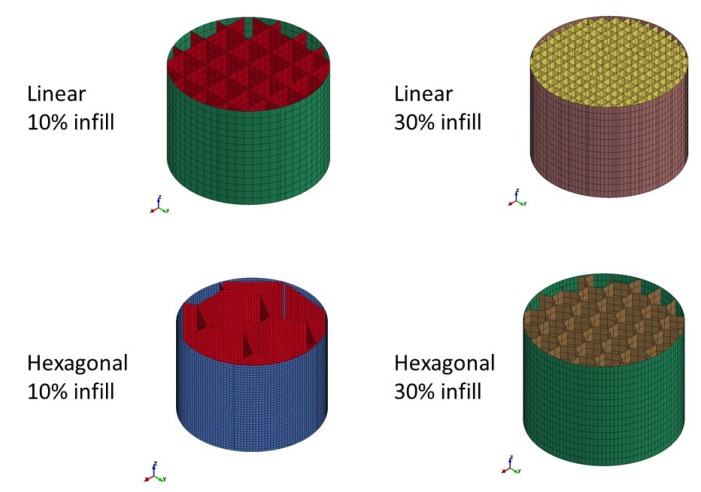
Isomeric views of virtual 3D printed specimen sections.

**Figure 14 materials-13-00457-f014:**
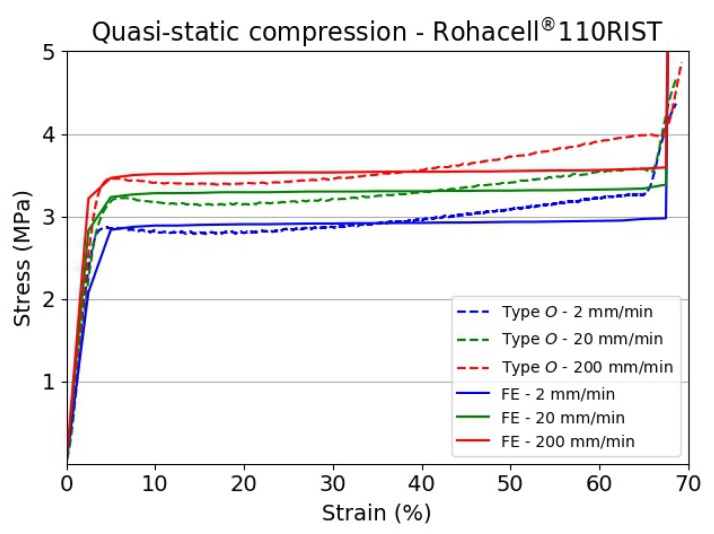
Comparison between numerical and experimental results for specimen Type *O* (Rohacell^®^ 110RIST).

**Figure 15 materials-13-00457-f015:**
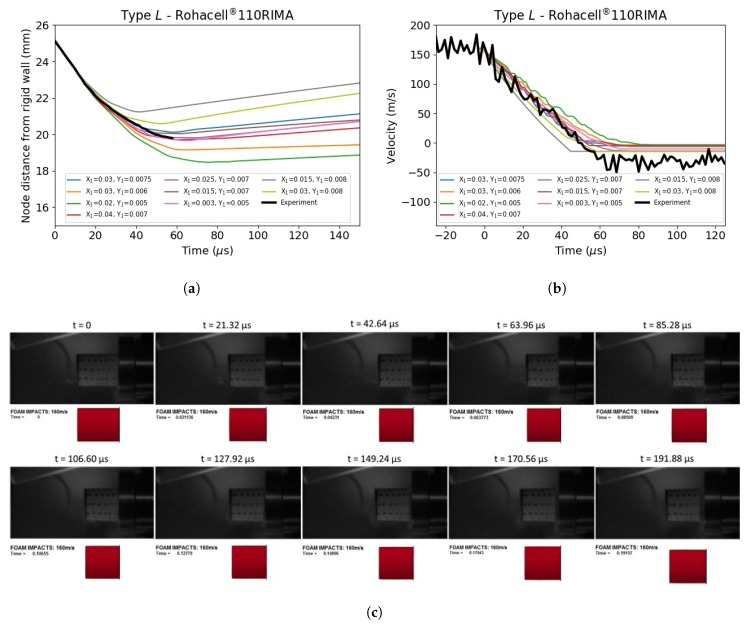
FE study for Rohacell^®^ 110RIMA. (**a**) Crash distance vs. time; (**b**) velocity vs. time; (**c**) comparison between experimental and FE results for X_1_ = 0.025 and Y_1_ = 0.007.

**Figure 16 materials-13-00457-f016:**
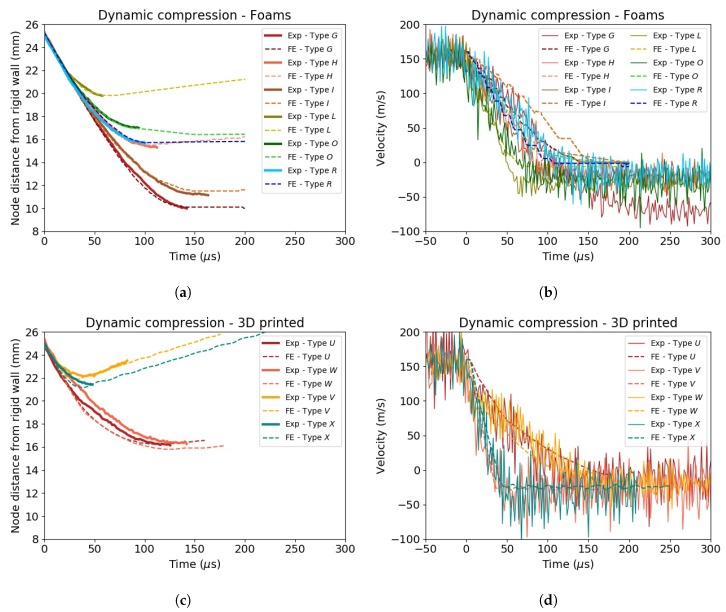
Comparison between experimental and FE results for foams and 3D printed specimens. (**a**,**c**) Crash distance vs. time; (**b**,**d**) velocity vs. time.

**Figure 17 materials-13-00457-f017:**
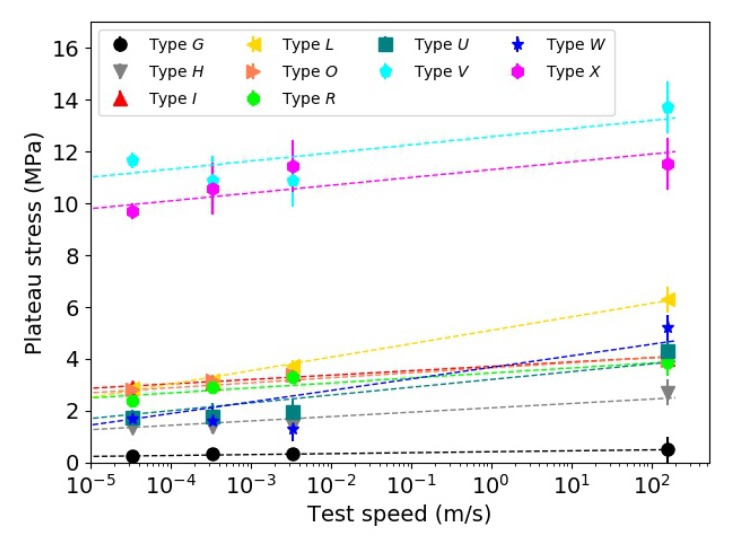
Plateau stress vs. test speed for the investigated materials.

**Table 1 materials-13-00457-t001:** Physical properties of the investigated materials: foams.

Trade Name	Material	Density kg/m${}^{3}$	ID Code	Dynamic Tests
N/A	EPP	30	*A*
N/A	EPP	60	*B*
N/A	EPP	90	*C*
N/A	EPS	50	*D*
N/A	EPS	110	*E*
Plastazote^®^ HD30	PE	30	*F*
Plastazote^®^ HD60	PE	60	*G*	Yes
N/A	PET	115	*H*	Yes
N/A	PET	180	*I*	Yes
Rohacell^®^ 51RIMA	PMI	51	*J*
Rohacell^®^ 71RIMA	PMI	71	*K*
Rohacell^®^ 110RIMA	PMI	110	*L*	Yes
Rohacell^®^ 51RIST	PMI	51	*M*
Rohacell^®^ 71RIST	PMI	71	*N*
Rohacell^®^ 110RIST	PMI	110	*O*	Yes
Rohacell^®^ 51IG	PMI	51	*P*
Rohacell^®^ 71IG	PMI	71	*Q*
Rohacell^®^ 110IG	PMI	110	*R*	Yes
Zotek^®^ F30	PVDF	30	*S*
Zotek^®^ F74	PVDF	74	*T*

**Table 2 materials-13-00457-t002:** Physical properties of the investigated materials: 3D printed.

Raw Material Brand	Material	Structure	Density kg/m^3^	ID Code	Dynamic Tests
RS-Pro	ABS	10% hexagonal fill	240	*U*	Yes
RS-Pro	ABS	30% hexagonal fill	392	*V*	Yes
RS-Pro	ABS	10% linear fill	240	*W*	Yes
RS-Pro	ABS	30% linear fill	427	*X*	Yes
Ultimaker	CPE	10% linear fill	200	*Y*	
Ultimaker	CPE	30% linear fill	315	*Z*	
Ultimaker	CPE	10% triangular fill	193	*AA*	
Ultimaker	CPE	30% triangular fill	314	*AB*	
Ultimaker	PA	10% linear fill	226	*AC*	
Ultimaker	PA	30% linear fill	425	*AD*	
Ultimaker	PA	10% triangular fill	216	*AE*	
Ultimaker	PA	30% triangular fill	414	*AF*	
Ultimaker	PLA	10% linear fill	284	*AG*	
Ultimaker	PLA	30% linear fill	518	*AH*	
Ultimaker	PLA	10% triangular fill	273	*AI*	
Ultimaker	PLA	30% triangular fill	509	*AJ*	

**Table 3 materials-13-00457-t003:** Settings used for the preparation of 3D printed specimens.

3D Printer	Material	Temperature		No. Shells Infill/Skin	Layer Height (mm)
		Built Plate (°C)	Nozzle (°C)		
Makerbot Replicator 2x	ABS	110	230	1/2	0.3
Ultimaker 3 Extended	CPE	80	235	1/1	0.4
Ultimaker 3 Extended	PA	60	245	1/1	0.4
Ultimaker 3 Extended	PLA	60	195	1/1	0.4

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
