# Peer review of "On the Compressive Response of Polymeric Cellular Materials"

_materials, 2020, doi:10.3390/ma13020457_

Round 1

Reviewer 1 Report

I found no factual errors. The work should be published in  POLYMERS.

Abbreviations used in this manuscript should be at the beginning.

Author Response

I found no factual errors. The work should be published in POLYMERS.

Thank you for your comment. In the authors’ opinion, the scope of Materials journal is more pertinent with the work and analysis presented in the manuscript. Polymers focusses predominantly on the synthesis, analysis and physics of the polymers, with less interest on their macromechanical properties.

Abbreviations used in this manuscript should be at the beginning.

Thank you for your comment. The manuscript has been drafted according to the Journal’s template, which presents the ‘Abbreviations’ section at the end of the paper. The editing team will modify the style if necessary.

Reviewer 2 Report

The article analysis various 3D printed polymeric cellular materials and their mechanical response. I think article is quite interesting but I have some remarks which I hope will be considered.

Firstly, I do not think that photos of the test methods are necessary (Fig. 1 and Fig.2). Instead, it would be nice to see photos of 3D printing process if possible.

I suggest transferring Fig. 6 to Materials and Methods part.

Check the first references in your Reference list, they are missing.

Author Response

The article analysis various 3D printed polymeric cellular materials and their mechanical response. I think article is quite interesting but I have some remarks which I hope will be considered.
Firstly, I do not think that photos of the test methods are necessary (Fig. 1 and Fig.2). Instead, it would be nice to see photos of 3D printing process if possible.

Thank you for your comment. In the authors’ opinion, it is necessary to show Figure 1 and Figure 2 to understand the differences in the raster and internal topology of 3D printed structures and the testing setup, respectively. The 3D printing outcome is in Figure 1. Describing the actual 3D printing process is not in the scope of the manuscript hence not shown.

I suggest transferring Fig. 6 to Materials and Methods part.

Thank you for your comment. Figure 6 should be placed in the ‘Results and Discussion’ section as it shows the specimens before and after testing.

Check the first references in your Reference list, they are missing.
Thank you for your comment. References updated.

Reviewer 3 Report

In this paper authors reported the experimental results on “On the compressive response of polymeric cellular materials”. The writing of the article is superb. The presentation of data and experimentation is outstanding. I have a few points to make more readable for the audience.

The x-axis of Figure 8 should be modified. It is not clear. Figure 17, the values on XY- axis are not clear. Increase the size of the text. REF [4], [13] and [34] don’t look consistent with other references. Please, make all References consistent.

Good work. Congratulations. 

Author Response

In this paper authors reported the experimental results on “On the compressive response of polymeric cellular materials”. The writing of the article is superb. The presentation of data and experimentation is outstanding. I have a few points to make more readable for the audience.

The x-axis of Figure 8 should be modified. It is not clear.

Thank you for your comment. Figure 8 updated.

Figure 17, the values on XY- axis are not clear. Increase the size of the text.

Thank you for your comment. Figure 17 updated.

REF [4], [13] and [34] don’t look consistent with other references. Please, make all References consistent.

Thank you for your comment. The reference of books would have a different style compared to that of journal papers. The editing team will modify the style if necessary.

Good work. Congratulations.

Round 2

Reviewer 2 Report

I don't know it is, but I think that article title is misleading or something wrong with it. Please correct it.

Author Response

I don't know it is, but I think that article title is misleading or something wrong with it. Please correct it.

Thank you for your comment. The authors believe that the title of the manuscript should be left as it is because it states and summarises the main objective of the study.